# Mode-Based Analysis and Optimal Operation of MSF Desalination System

**Hanhan Gao, Aipeng Jiang \*, Qiuyun Huang, Yudong Xia, Farong Gao and Jian Wang**

School of Automation, Hangzhou Dianzi University, Hangzhou 310018, China; wh_isper@163.com (H.G.); hqy1997@hdu.edu.cn (Q.H.); ydxia@hdu.edu.cn (Y.X.); frgao@hdu.edu.cn (F.G.); wj@hdu.edu.cn (J.W.)

\* Correspondence: jiangaipeng@hdu.edu.cn

**Abstract:** Multi-stage flash (MSF) desalination plays an important role in achieving large-scale fresh water driven by thermal energy. In this paper, based on first-principle modeling of a typical multi-stage flash desalination system, the effects of different operational parameters on system performance and operational optimization for cost saving were extensively studied. Firstly, the modelled desalination system was divided into flash chamber modules, brine heater modules, mixed modules and split modules, and based on energy and mass conservation laws the equations were formulated and put together to describe the whole process model. Then, with physical parameter calculation the whole process was simulated and analyzed on the platform of MATLAB, and the water production performance effected by operational parameters such as the feed temperature of seawater, the recycle brine from the discharge section, steam temperature and flowrate of recycled brine were discussed and analyzed. Then, the optimal operation to achieve maximize GOR (gained output ratio) with fixed freshwater demand was considered and performed, and thus the optimal flowrate of recycled brine, steam temperature, and seawater output flowrate from rejection section were obtained based on the established model. Finally, considering that minimizing the daily operational cost is a more rational objective, the operational cost equations were formulated and the optimal problem to minimize the daily operational cost was solved and the optimal manipulated variables at different hours were obtained. The study results can be used for guideline of real time optimization of the MSF system.

**Keywords:** MSF; seawater desalination; simulation; optimal operation; first principle modeling

## 1. Introduction

In the present society, the shortage of fresh water resources is an indisputable fact. As a necessity of human life, the lack of fresh water resources have become a constraint on social development in many countries of the world [1,2]. With the development of industrial technology and the world population quick increase, the global demand for desalination is increasing [3,4]. At the present time, desalination plants produce around 95 million $m^3$/day, Middle East and North Africa (MENA) regions are responsible for 48% of the global installed desalination capacity [5,6]. Generally, the use of multi-stage flash (MSF), multi-effect distillation (MED), electrodialysis (ED), and reverse osmosis (RO) processes has attracted significant attention in attempts to improve the reliability and the performance of freshwater production processes. Among them, MSF desalination is an important technology in the desalination industry due to its high reliability and good distillate quality. However, it is acknowledged that its production cost is high and the system performance is greatly affected by the seawater temperature, fouling factor and etc. [7–11]. Therefore, it is of great significance to study how to reduce the operating cost of the MSF desalination system. In order to solve this problem, it is necessary to establish a complete mathematical model.

In past decades, the simulation, performance analysis and optimization of MSF systems were widely performed to reveal the desalination mechanism and to improve the performance with lower cost. Hellal [12] et al. established a detailed mathematical model of MSF desalination system, and then proposed an efficient and reliable method to solve the nonlinear equations describing the multi-stage flash desalination system. After linearization, the equations are decomposed into some subsets based on their characters, and then the enthalpy balance equations are expressed in the form of triangular matrix. This method has good stability and fast convergence speed, but the heat loss of the flash process is ignored in the model. Therefore, Marina Rosso [13] et al. established a detailed steady-state mathematical model for the analysis of MSF desalination process. The model takes into account the physical and structural factors, such as the geometry of each flash chamber, the changes in physical properties of water with temperature, non-equilibrium temperature difference and so on. The established mathematical model allows the analysis of the influence of operation and design variables on the system in the process. This information can be used not only for design purposes but also for the development of dynamic models. Based on the above model, El-Dessouky [14] further considered the factors, including fouling factor, pressure drop through demister, non-equilibrium allowance and so on into the model, making it stricter and more accurate. Considering that the parameters such as the boiling point elevation, heat capacity of brine, latent heat of vaporization are functions of temperature and salinity, and the non-equilibrium loss in the MSF process should not be ignored, Wu [15] et al. established a mathematical model of the MSF desalination system. However, in order to reduce the non-linearity of the system, the temperature of each flash chamber is assumed to follow an equal temperature drop distribution, which reduced the accuracy the rationality of the modeling process. Therefore, Tanvir and Mujtaba [16] proposed an effective prediction method of boiling point elevation of brine based on neural network, and then Tanvir and Mujtaba [17] embedded this method into the MSF desalination process model written in gPROMS platform. Based on the above work, Tanvir and Mujtaba studied the influence of seawater temperature and steam temperature on the performance of the MSF system. Since brine heater fouling can significantly reduce the heat transformation efficiency and cause performance deterioration of the MSF system, Al-Rawajfeh et al. [18] studied the deposition of calcium carbonate in flash chambers in once-through MSF (MSF-OT) and brine recirculation MSF (MSF-BR) processes by correlating the deposition of calcium carbonate to the released rate of carbon dioxide in a steady state model based on coupling of mass transfer with chemical reaction. Then they extended their work to include deposition of calcium sulphate with calcium carbonate inside the tubes and flash chambers [19]. Mujtaba considered the brine heater fouling in his MSF process simulation and optimization. Based on actual plant data, a simple linear dynamic fouling factor profile was developed which allows calculation of the fouling factor at different times [20]. Furthermore, Alsadaie and Mujtaba [10] presented very detailed fouling model that considered the attached and removal rate of calcium carbonate and magnesium hydroxide and also it considered the effect of temperature, velocity and salinity. The model was applied on the MSF-BR process. In 2019, they presented a dynamic model of fouling to predict the crystallization of calcium carbonate and magnesium hydroxide inside the condenser tubes of the once-through MSF process [21].

Based on the mechanism analysis and mathematical modeling of MSF desalination, Hu et al. [22] developed an improved optimization model for the MSF desalination system using the annual average distillate cost as the objective function, and a more detailed optimization design analysis with market price change was carried out for a 3000-ton/day MSF desalination plant. Based on the strict mathematical model of MSF system, Mussati et al. [23] studied the design optimization model to obtain a better structure of an MSF system, and generalized gradient algorithm under the GAMS platform was used to solve the problem. Said et al. [24] considered a storage tank between the desalination system and the users, then they built the design optimization and operational optimization problem to minimize the daily operational cost of the MSF desalination system under various conditions. Hawaidi and Mujtaba [20] considered the dynamic relationship between scaling factor of the brine heater, operation

time and seawater temperature change, they also studied the operational optimization problem with the goal of minimum annual operation cost with a fixed freshwater demand. Considering the dynamic change of operational conditions and the aim of better control of a MSF system, dynamic modeling and optimization of the MSF process were also studied in past years. Based on a detailed description of the fundamental elementary phenomena involved in the process, Mazzotti et al. [25] developed a model for the dynamic simulation of MSF desalination units and analyzed the non-linear dynamic features under different disturbances. Gambier et al. [26] collected some different dynamic models from the literatures, and analyzed their advantages and drawbacks taking into account simulation and automatic control purposes. Considering that there were not enough detailed models and analyses of the dynamics of the MSF process and the demister, Al-Fulaij [27] developed lumped parameter dynamic models for the once-through (MSF-OT) and the brine circulation (MSF-BC) processes. He also coded the models with the gPROMS modelling program for analysis. By coupling the dynamic equations of mass, energy and momentum, Bodalal et al. [28] presented a mathematical model to predict the performance of MSF plant systems under transient conditions. The model describing the dynamic behavior of each stage in terms of some key physical parameters were solved by using the fifth order Runge–Kutta method. Lappalainen et al. [29] presented a new method for one-dimensional modelling and dynamic simulation of thermal desalination processes. The approach combines the simultaneous mass, momentum, and energy solution, local phase equilibrium by Rachford-Rice equation, and rigorous calculation of the seawater properties as function of temperature, pressure and salinity. Computing results show that it is a competent approach for dynamic simulation of thermal desalination processes. With known dynamics of the MSF process, advanced control can be obtained for performance improvement and cost saving. Alsadaie and Mujtaba [30] coded a dynamic model of MSF process with gPROMS model builder, and successfully applied a generic model control (GMC) algorithm to provide better performance over a conventional PID (proportional-integral-derivative) controller. Their work has guiding significance for the optimal operation and control of MSF process in the short term to long term, and is very helpful and enlightening to our work.

The modelling and optimization studies mentioned above established the solid foundation for reliable for economic operation of MSF system. But few of them considered the effect of discharged recycle mass flow and pump energy cost for the whole flowsheet. We also found that for the easy use and solution of the MSF process model, ideal assumptions and some heat loss were ignored, and the components of operating expenses are quite different if we consider the power loss of pumps and steam price. Therefore, in order to obtain a more complete and elaborated model, this paper considers the effects of factors such as boiling point elevation, influence of heat loss of condenser tubes, the phenomenon of unequal temperature drop in each flash stage, and at the same time the compensation effect of the reject seawater recycle mass flowrate on the seawater temperature is also accounted for in the proposed model. Since we just consider daily operational optimization of the MSF process, a fouling factor was used in the process model, and the dynamic model of exchanger fouling was not incorporated into the established model. The established model can fully reflect the non-linearity of the system, and can reflect the influence of various operating variables on the system performance well in short term. Based on the model, the operating states of the MSF system were simulated under different parameter conditions. And in addition, considering the changes of seawater temperature and distillate demand on the system performance, the optimal operation problems were carried out for maximizing the GOR (gained output ratio) and minimizing the daily operational cost, respectively.

## 2. Flowsheet of Multi-Stage Flash (MSF) Process

MSF desalination system has two classic structures, which are one-through multi-stage flash (OT-MSF) seawater desalination system and brine recirculation multi-stage flash (BR-MSF) seawater desalination system [27,30–33]. Among them, the structure of OT-MSF system is relatively simple, and the brine circulation type is more widely used because of its better comprehensive performance.

The flowsheet of BR-MSF system is shown in Figure 1. The simulation and optimization research of this paper are based on this kind of BR-MSF system.

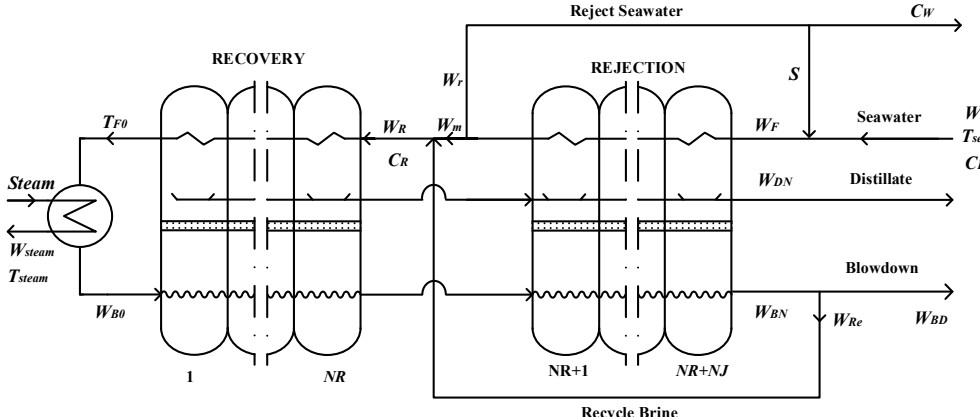

**Figure 1.** Multi-stage flash (MSF) desalination system.

The BR-MSF system is composed of four parts: brine heater, heat rejection section, heat recovery section, splitters and mixers. The heat rejection section aims to reject the remaining heat energy from the system, thereby cooling the distilled product and concentrated brine to the lowest possible temperature. In addition to completing the energy release from the heat rejection section, the splitters and mixers also recycle part of the waste brine back into the system to reuse the energy.

As can be seen from Figure 1, after preheating of the heat rejection section, the feed seawater is divided into two parts, one is returned to the sea and the other is mixed with the recycled brine. The mixed brine is pumped into the end of the heat recovery section. When it flows through a series of heat exchangers from the right to the left, it is gradually heated, and the flash steam in the flash chamber is condensed. Finally, seawater comes out of the first stage flash chamber of the heat recovery section, flows into the brine heater to be further heated, and then flows into the first stage flash chamber again at the highest temperature. The pressure in the flash chamber is gradually reduced, so that the flash stream enters the flash chamber and evaporates immediately. The generated steam passes through the condensers and drops into the distillate trays after being condensed. The process is repeated until the last stage is reached, then the concentrated brine is rejected, the distillate is extracted, and a part of the brine is reused as recycle brine.

## 3. Mathematical Model of MSF Process

According to the relationship of energy and momentum conservation laws and overall mass balance, a steady state model of the MSF system can be established based on the following assumptions:

(1)  The distillate from whatever stage is salt free;
(2)  Non-condensable gases are ignored;
(3)  The system is adiabatic.

The complete mathematical model of MSF desalination system includes four parts: flash chamber module, brine heater module, splitters and mixers module and physical parameter equations.

### 3.1. Flash Chamber Module

For the *j*-th stage flash chamber, the relationship of each flowrate in the flash process is shown in Figure 2, and the flash chamber model is established according to the relationship of flowrate.

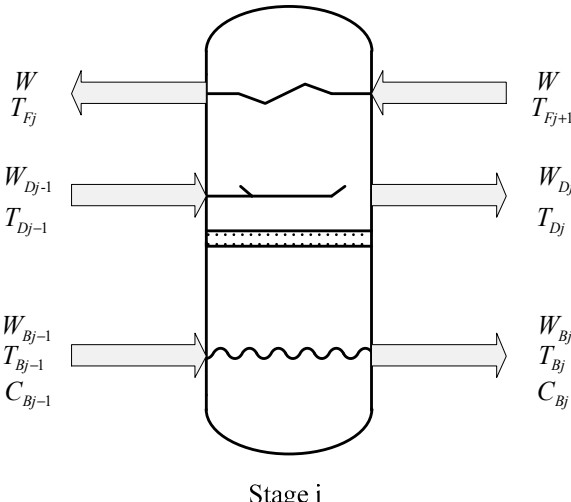

**Figure 2.** Flowrate relations of the *j*-th stage flash chamber in an MSF system.

Overall mass balance:

$$W_{Bj-1} + W_{Dj-1} = W_{Bj} + W_{Dj} \tag{1}$$

Salt mass balance:

$$W_{Bj-1}C_{Bj-1} = W_{Bj}C_{Bj} \tag{2}$$

Enthalpy balance of flashing brine:

$$W_{Bj-1}h_{Bj-1} = W_{Bj}h_{Bj} + V_{Bj}h_{Vj} \tag{3}$$

$$W_{Bj-1} - W_{Bj} = V_{Bj} \tag{4}$$

Overall enthalpy balance:

$$\begin{aligned} WCP_{Rj}(T_{Fj} - T_{Fj+1}) \quad = \quad & W_{Dj-1}CP_{Dj-1}(T_{Dj-1} - T^*) + W_{Bj-1}CP_{Bj-1}(T_{Bj-1} - T^*) \\ & -W_{Dj}CP_{Dj}(T_{Dj} - T^*) - W_{Bj}CP_{Bj}(T_{Bj} - T^*) \end{aligned} \tag{5}$$

Heat transfer equation:

$$WCP_{Rj}(T_{Fj} - T_{Fj+1}) = \frac{U_j A_j(T_{Fj} - T_{Fj+1})}{\ln((T_{Dj} - T_{Fj+1})/(T_{Dj} - T_{Fj}))} \tag{6}$$

where, $U_j = \phi(W, T_{Fj}, T_{Fj+1}, T_{Dj}, D_j^i, D_j^0, f_j)$.

$W$ in Equations (4) and (5) refer to the feed flowrate of the *j*-th stage flash chamber. For the heat rejection section, $W = W_F$ represents to the feed seawater flowrate, while for the heat recovery section, $W = W_R$ represents to the feed stream flowrate of the heat recovery section.

Temperature relationship:

$$T_{Bj} = T_{Dj} + \Delta BPE_j + \Delta NETD_j + \Delta TL_j \tag{7}$$

$$T_{Vj} = T_{Dj} + \Delta TL_j \tag{8}$$

*3.2. Brine Heater*

Overall mass balance:

$$W_{B0} = W_R \tag{9}$$

Salt mass balance:

$$C_{B0} = C_R \tag{10}$$

Overall enthalpy balance:

$$W_R CP_{RH}(T_{B0} - T_{F1}) = W_{steam}\lambda_s \tag{11}$$

where, $\lambda_s = 597.9541 - 0.5753T_{steam} + 0.2849 \times 10^{-3}T_{steam}^2 - 0.3791 \times 10^{-5}T_{steam}^3$.

Heat transfer equation:

$$W_R CP_{RH}(T_{B0} - T_{F1}) = \frac{U_H A_H(T_{B0} - T_{F1})}{\ln((T_{steam} - T_{F1})/(T_{steam} - T_{B0}))} \tag{12}$$

where, $U_H = \phi(W, T_{B0}, T_{F1}, T_{steam}, D_H^i, D_H^0, f_{BH})$.

## 3.3. Splitters and Mixers Module

Splitters:

$$W_{BD} = W_{BN} - W_{Re} \tag{13}$$

$$W_m = W_F - W_r \tag{14}$$

$$S = W_r - C_w \tag{15}$$

Mixers:

$$W_R = W_{Re} + W_m \tag{16}$$

$$W_R \cdot C_R = W_{Re} \cdot C_{Re} + W_m \cdot C_m \tag{17}$$

$$W_R \cdot h_R = W_{Re} \cdot h_{Re} + W_m \cdot h_m \tag{18}$$

$$W_F = S + W_S \tag{19}$$

$$W_F \cdot h_{W_F} = S \cdot h_S + W_S \cdot h_{W_S} \tag{20}$$

## 3.4. Physical Parameter Equations

Multi-stage flash desalination system exhibits a strong nonlinearity. In order to achieve a better simulation performance, the physical and chemical properties of the model variables in the system must be well characterized. The physical parameter equations in this paper refer to the research of Woldai et al. [34].

Specific heat capacity calculation equation:

$$\begin{aligned} CP_D &= 1.001183 - 6.1666652 \times 10^{-5}T_D + 1.3999989 \times 10^{-7}T_D^2 \\ &\quad + 1.3333336 \times 10^{-9}T_D^3 \end{aligned} \tag{21}$$

$$CP_B = [1 - CB(0.011311 - 1.146 \times 10^{-5}T_B)] \times CP_D \tag{22}$$

Enthalpy calculation equation:

$$W_R = W_{Re} + W_m \tag{23}$$

$$h_B = CP_B \cdot T_B \tag{24}$$

$$h_D = CP_D \cdot T_D \tag{25}$$

Calculation equation of flash temperature difference loss:

$$
\begin{aligned}
BPE \;=\; & C \cdot T^2 / (266919.6 - 379.669T + 0.334169T^2) \\
& \times [565.757/T - 9.81559 + 1.54739 \ln T \\
& - C(337.178/T - 6.41981 + 0.922753 \ln T) \\
& + C^2(32.681/T - 0.55368 + 0.079022 \ln T)]
\end{aligned}
\tag{26}
$$

where, $C = (19.819C_B)/(1 - C_B)$.

$$
NETD = \frac{195.556 \times (H_j/0.0254)^{1.1}(\omega_j \times 0.6706 \times 10^{-3})^{0.5}}{(1.8 \times \Delta T_B + 32)^{0.25} T_s^{2.5}}
\tag{27}
$$

where, $\omega_j = WF/w_j$.

$$
TL = exp(1.885 - 0.02063T_D)/1.8
\tag{28}
$$

Heat-transfer coefficient:

$$
U = 4.8857/(y + z + 4.8857f)
\tag{29}
$$

$$
y = [0.0013(v \times D^i)^{0.2}]/[(0.2018 + 0.0031 \times T)v]
\tag{30}
$$

$$
\begin{aligned}
z \;=\; & 0.1024768 \times 10^{-2} - 0.7473939 \times 10^{-5} T_D \\
& + 0.999077 \times 10^{-7} T^2{}_D - 0.430046 \times 10^{-9} T^3{}_D \\
& + 0.6206744 \times 10^{-12} T^4{}_D
\end{aligned}
\tag{31}
$$

## 4. Simulation and Analysis

### 4.1. Steady-State Simulation of MSF Desalination System

In the MSF desalination process, the gained output ratio (GOR) is a key indicator demonstrating the performance of the system [35], which is defined as follows:

$$
GOR = W_{DN}/W_{steam}
\tag{32}
$$

According to the MSF desalination process shown in Figure 2, the simulation is conducted using the developed steady-state model. Using the experimental data generated form the MSF system reported in Rosso's study [8], the proposed model could be further validated. The heat rejection section of the MSF system consists of a 3-stage flash chamber, and the heat recovery section consists of a 13-stage flash chamber. Relevant parameters of brine heater and flash chamber are shown in Tables 1 and 2.

**Table 1.** Brine heater parameters of the MSF system.

| Parameters | Unit | Numerical Value |
|---|---|---|
| Internal diameter of condenser ($D_H^i$) | m | 0.0220 |
| External diameter of condenser ($D_H^o$) | m | 0.0244 |
| Length of condenser ($L_H$) | m | 12.2 |
| Heat transfer area ($A_H$) | m$^2$ | 3530 |
| Fouling factor ($f_H$) | (h·m$^2$·K)/kcal | $1.86 \times 10^{-4}$ |

**Table 2.** Parameters of flash chamber of the MSF system.

| Parameters | Unit | Heat Recovery Section | Heat Rejection Section |
|---|---|---|---|
| Internal diameter of condenser ($D_j^i$) | m | 0.0220 | 0.0239 |
| Length of condenser ($L_j$) | m | 12.2 | 10.7 |
| Heat transfer area ($A_j$) | m$^2$ | 3995 | 3530 |
| Width of flash chamber ($w_j$) | m | 12.2 | 10.7 |
| Fouling factor ($f_j$) | (h·m$^2$·K)/kcal | $1.4 \times 10^4$ | $2.33 \times 10^5$ |

During the simulation, the seawater temperature ($T_{sea}$) and salt concentration ($C_F$) are set as 35 °C and 5.7%, respectively. The steam temperature ($T_{steam}$) is set as 97 °C, the feed seawater mass flowrate ($W_S$) is $11.3 \times 10^6$ kg/h, and rejected seawater mass flowrate ($C_W$) and recycle stream mass flowrate ($W_{Re}$) are $5.62 \times 10^6$ kg/h and $6.35 \times 10^6$ kg/h, respectively. Based on the above relevant parameters and operating conditions, a steady-state simulation of the seawater desalination process is performed. The simulation results are shown in Table 3, and the simulation results in the literature (in italics) are also listed in the table for comparison. From the calculation, the stream mass flowrate, distillate mass flowrate, the temperature distribution of each flash chamber, the concentration of brine and the steam consumption are obtained. In addition, the evaporation capacity of brine and the temperature of flash steam in each stage are also calculated. In order to make the data in the Table 3 clearer, our simulation results are placed in the first line, and the results of Rosso et al. are placed on the second line (in italics and gray color). From the Table 3, it can be seen that our simulation results agree well with the previous study of Rosso et al.

**Table 3.** Simulation results.

| Stage j | $W_{Bj}$ (kg/h) | $W_{Dj}$ (kg/h) | $C_{Bj}$ (wt%) | $T_{Fj}$ (°C) | $T_{Dj}$ (°C) | $T_{Bj}$ (°C) | $V_{Bj}$ (kg/h) | $T_{vj}$ (°C) |
|---|---|---|---|---|---|---|---|---|
| 0 | $1.2030 \times 10^7$ | | 6.2964 | 83.2410 | | 89.6610 | | |
| | *$1.203 \times 10^7$* | | *6.2922* | *83.33* | | *89.74* | | |
| 1 | $1.1969 \times 10^7$ | $6.0530 \times 10^4$ | 6.3282 | 80.3232 | 85.6664 | 86.8504 | 60,530.13 | 85.7450 |
| | *$1.197 \times 10^7$* | *$5.940 \times 10^4$* | *6.3234* | *80.41* | *85.75* | *86.89* | | |
| 2 | $1.1909 \times 10^7$ | $1.2148 \times 10^5$ | 6.3606 | 77.3566 | 82.7921 | 83.9859 | 60,952.13 | 82.8795 |
| | *$1.191 \times 10^7$* | *$1.187 \times 10^5$* | *6.3549* | *77.44* | *82.87* | *84.01* | | |
| 3 | $1.1847 \times 10^7$ | $1.8275 \times 10^5$ | 6.3935 | 74.3456 | 79.8655 | 81.0715 | 61,269.87 | 79.9629 |
| | *$1.185 \times 10^7$* | *$1.784 \times 10^5$* | *6.3869* | *74.43* | *79.95* | *81.08* | | |
| 4 | $1.1786 \times 10^7$ | $2.4423 \times 10^5$ | 6.4269 | 71.2944 | 76.8904 | 78.1116 | 61,481.50 | 76.9992 |
| | *$1.179 \times 10^7$* | *$2.385 \times 10^5$* | *6.4195* | *71.37* | *76.97* | *78.11* | | |
| 5 | $1.1724 \times 10^7$ | $3.0582 \times 10^5$ | 6.4606 | 68.2080 | 73.8712 | 75.1111 | 61,584.98 | 73.9929 |
| | *$1.173 \times 10^7$* | *$2.989 \times 10^5$* | *6.4525* | *68.28* | *73.94* | *75.09* | | |
| 6 | $1.1663 \times 10^7$ | $3.6740 \times 10^5$ | 6.4948 | 65.0914 | 70.8124 | 72.0749 | 61,577.98 | 70.9487 |
| | *$1.167 \times 10^7$* | *$3.595 \times 10^5$* | *6.4860* | *65.16* | *70.88* | *72.04* | | |
| 7 | $1.1601 \times 10^7$ | $4.2885 \times 10^5$ | 6.5292 | 61.9501 | 67.7189 | 69.0088 | 61,457.69 | 67.8718 |
| | *$1.161 \times 10^7$* | *$4.201 \times 10^5$* | *6.5198* | *62.01* | *67.78* | *68.95* | | |
| 8 | $1.1540 \times 10^7$ | $4.9007 \times 10^5$ | 6.5638 | 58.7898 | 64.5959 | 65.9185 | 61,220.58 | 64.7677 |
| | *$1.155 \times 10^7$* | *$4.806 \times 10^5$* | *6.5540* | *58.84* | *64.65* | *65.84* | | |
| 9 | $1.1479 \times 10^7$ | $5.5094 \times 10^5$ | 6.5986 | 55.6168 | 61.4491 | 62.8108 | 60,862.28 | 61.6421 |
| | *$1.149 \times 10^7$* | *$5.410 \times 10^5$* | *6.5885* | *55.65* | *61.49* | *62.70* | | |
| 10 | $1.1419 \times 10^7$ | $6.1131 \times 10^5$ | 6.6335 | 52.4379 | 58.2844 | 59.6925 | 60,377.35 | 58.5015 |
| | *$1.143 \times 10^7$* | *$6.010 \times 10^5$* | *6.6231* | *52.46* | *58.32* | *59.55* | | |
| 11 | $1.1359 \times 10^7$ | $6.7107 \times 10^5$ | 6.6684 | 49.2603 | 55.1084 | 56.5715 | 59,759.11 | 55.3527 |
| | *$1.137 \times 10^7$* | *$6.606 \times 10^5$* | *6.6578* | *49.27* | *55.13* | *56.39* | | |
| 12 | $1.1300 \times 10^7$ | $7.3007 \times 10^5$ | 6.7032 | 46.0923 | 51.9280 | 53.4562 | 58,999.50 | 52.2029 |
| | *$1.131 \times 10^7$* | *$7.197 \times 10^5$* | *6.6925* | *46.09* | *51.93* | *53.24* | | |
| 13 | $1.1242 \times 10^7$ | $7.8816 \times 10^5$ | 6.7378 | 43.8680 | 48.7509 | 50.3560 | 58,088.83 | 49.0603 |
| | *$1.125 \times 10^7$* | *$7.780 \times 10^5$* | *6.7272* | *44.06* | *48.74* | *50.09* | | |
| 14 | $1.1192 \times 10^7$ | $8.3822 \times 10^5$ | 6.7680 | 40.9495 | 45.9537 | 47.6574 | 50,060.89 | 46.2969 |
| | *$1.120 \times 10^7$* | *$8.296 \times 10^5$* | *6.7582* | *41.10* | *45.87* | *47.28* | | |
| 15 | $1.1141 \times 10^7$ | $8.8884 \times 10^5$ | 6.7987 | 37.9936 | 43.1039 | 44.9038 | 50,620.84 | 43.4855 |
| | *$1.115 \times 10^7$* | *$8.816 \times 10^5$* | *6.7897* | *38.07* | *42.95* | *44.42* | | |
| 16 | $1.1090 \times 10^7$ | $9.3966 \times 10^5$ | 6.8299 | 35.0000 | 40.2000 | 42.1145 | 50,814.02 | 40.6250 |
| | *$1.110 \times 10^7$* | *$9.341 \times 10^5$* | *6.8219* | *35.00* | *39.98* | *41.51* | | |
| $W_{steam}$: | $1.3499 \times 10^5$ kg/h | | | *GOR:* | 6.96 | | | |
| | *$1.3489 \times 10^5$ kg/h* | | | | *6.92* | | | |

### 4.2. Analysis of MSF Desalination System

In order to further investigate the characteristics of the MSF desalination system based on the steady-state model of MSF desalination system established in this paper, more simulations were carried out to examine the influences of, the feed seawater temperature ($T_{sea}$), the reject seawater recycle mass flowrate ($S$), the steam temperature ($T_{steam}$), and the recycle stream mass flowrate ($W_{Re}$) on system performance. During the analysis, only the analyzed parameter was changed, all other parameters were fixed for the simulation, so as to obtain the system performances and state changes as the concerned parameter changes.

#### 4.2.1. Effect of Seawater Temperature on System Performance

The feed seawater temperature which varies all the time is an important variable in the process of the MSF desalination system. Its change will inevitably have an impact on the performance of the system [20,36]. In this paper, the seawater temperature fluctuation is simulated within the range of 5 °C–46 °C and the results are shown in Figures 3–5.

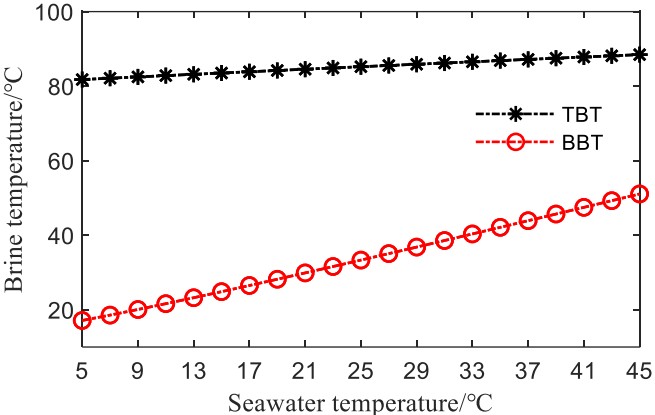

**Figure 3.** Effect of seawater temperature ($T_{sea}$) on brine temperature (top brine temperature (TBT) and bottom brine temperature (BBT)).

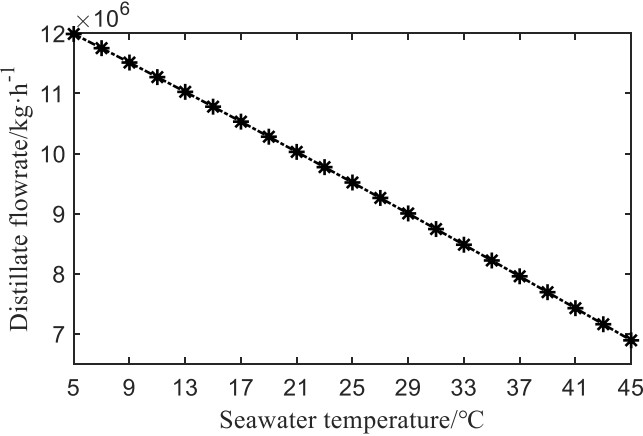

**Figure 4.** Effect of seawater temperature ($T_{sea}$) on distillate flowrate ($W_{DN}$).

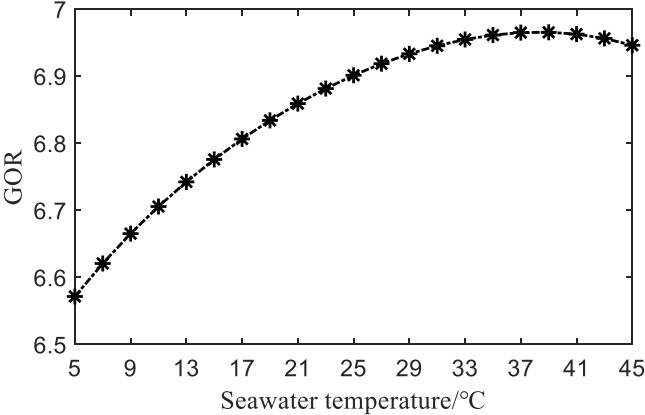

**Figure 5.** Effect of seawater temperature ($T_{sea}$) on gained output ratio (GOR).

As shown in Figure 3, the temperature of brine from the last flash chamber (BBT, the bottom brine temperature) and the temperature of brine at the outlet of the brine heater (TBT, the top brine temperature) can be obtained as the seawater temperature changes. As seen from Figure 3, BBT is sensitive to the change of seawater temperature ($T_{sea}$), while TBT remains almost unchanged. Thus, for the overall system, the temperature difference between the stages must be reduced when the number of flash chamber stages is constant. This situation will directly affect the distillate production ($W_{DN}$), as shown in Figure 4, as the temperature of the seawater ($T_{sea}$) increases, the distillate production ($W_{DN}$) decreases significantly.

Figure 5 shows the profile of GOR led by the change of seawater temperature ($T_{sea}$). It can be seen from Figure 5 that the GOR increases at first and then decreases with the increase of seawater temperature ($T_{sea}$). It is not too hard to understand that the seawater temperature ($T_{sea}$) increases and the TBT changes little, which means less steam consumption. From this perspective, GOR should show an increasing trend, and the decrease in distillate production ($W_{DN}$) will inevitably leads to the decrease of GOR. Since the change trend of GOR is related to specific system parameters, operating conditions and other factors, there should be a best feed seawater temperature ($T_{sea}$). As far as this system is concerned, the simulation result shows that the GOR presents the maximum value when the seawater temperature ($T_{sea}$) is 38 °C.

#### 4.2.2. Effect of Reject Recycle Mass Flowrate ($S$) on System Performance

It can be seen from the above analysis that when the seawater temperature ($T_{sea}$) is low, it will greatly affect the system performance. In winter or in the early morning when the seawater temperature ($T_{sea}$) is low, we can increase reject seawater recycle mass flowrate ($S$) for a certain temperature compensation to make the system more stable. In this case, the system performance was investigated as the reject recycle mass flowrate ($S$) changes in the range of $0$–$4 \times 10^6$ kg/h.

Increasing the reject recycle mass flowrate ($S$) actually increases the temperature of the feed seawater, so the change of reject recycle mass flowrate ($S$) is consistent with change of the feed seawater temperature ($T_{sea}$). But the effect is quite different, Figure 6 shows that BBT and TBT changes moderately as the reject recycle mass flowrate increases. Figure 7 shows if we increase the reject recycle mass flowrate ($S$) from 0 to $4 \times 10^6$ kg/h, the distillate flowrate ($W_{DN}$) reduced about 6.6%. This means if we want to maximize the distillate ($W_{DN}$), the reject recycle mass flowrate ($S$) should be set to zero.

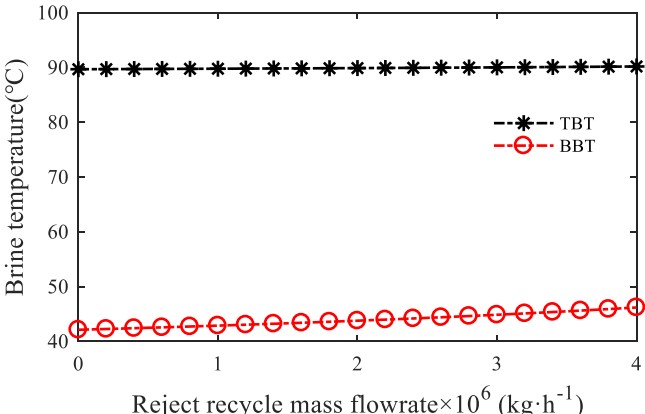

**Figure 6.** Effect of reject recycle mass flowrate (*S*) on brine temperature (TBT and BBT).

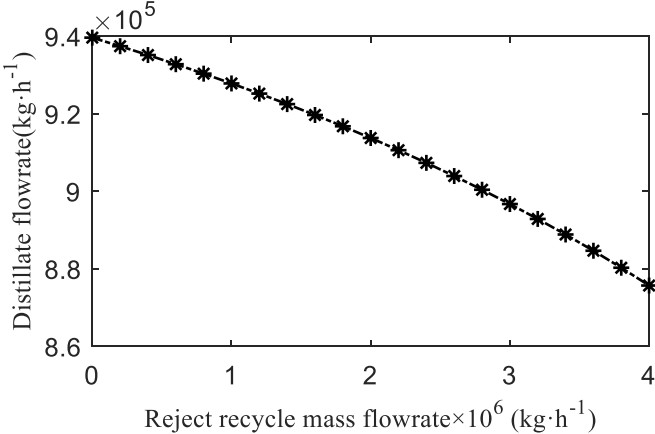

**Figure 7.** Effect of reject recycle mass flowrate (*S*) on distillate flowrate ($W_{DN}$).

### 4.2.3. Effect of Steam Temperature ($T_{steam}$) on System Performance

Steam releases latent heat ($\lambda_s$) to heat recycle brine ($W_{Re}$) in the system, so its temperature directly affects TBT, and thus the performance of the whole system. The performances of the system are simulated and analyzed when the temperature of steam ($T_{steam}$) is varied within the range of 90–120 °C. The simulation results are shown in Figures 8–10.

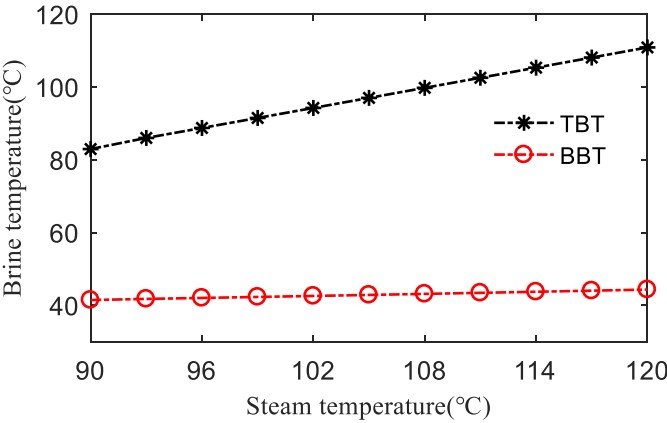

**Figure 8.** Effect of steam temperature ($T_{steam}$) on brine temperature (TBT and BBT).

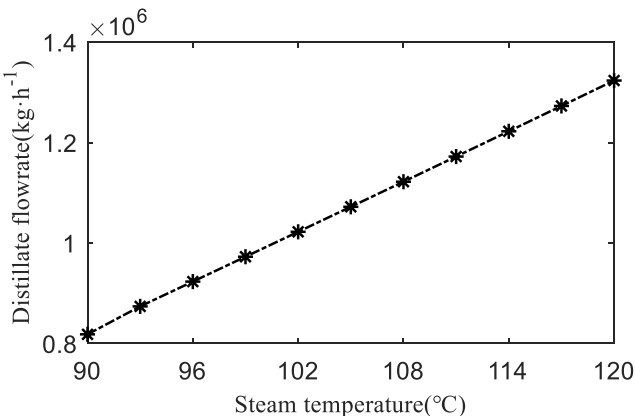

**Figure 9.** Effect of steam temperature ($T_{steam}$) on distillate flowrate ($W_{DN}$).

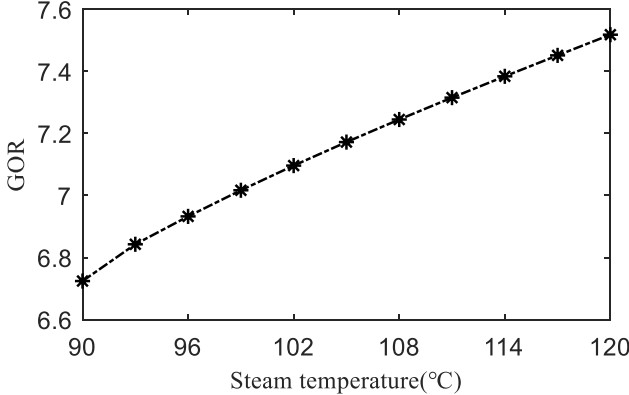

**Figure 10.** Effect of steam temperature ($T_{steam}$) on GOR.

It can be seen from Figure 8 that the change of steam temperature ($T_{steam}$) has a greater impact on TBT than that on BBT. Compared with Figure 3, it is not difficult to find that the sensitivity of TBT and BBT to the brine temperature and the steam temperature ($T_{steam}$) displays the opposite trend.

Figure 9 shows the effect of steam temperature on distillate flowrate. It can be seen that the distillate flowrate increases sharply with the increase of steam temperature. This is due to the increase in TBT and the increase in the total flash temperature difference can largely increase the evaporation capacity, and will cause the distillate production to increase accordingly.

It can be seen from Figure 10 that the change trend of GOR is directly proportional to the change trend of steam temperature ($T_{steam}$). From the perspective of thermodynamics, the higher the temperature of steam is, the less steam is needed for the same latent heat ($\lambda_s$). From this perspective, the GOR will increase. From the perspective of water production, the distillate production ($W_{DN}$) will increase with the rise of steam temperature, and thus the GOR. In summary, the increase of the steam temperature ($T_{steam}$) can improve the performance of the system, but the higher temperature steam ($T_{steam}$) also represents higher cost, which needs to be considered according to the actual situation.

### 4.2.4. Effect of Recycle Stream Flowrate ($W_{Re}$) on System Performance

Recycle stream flowrate ($W_{Re}$) is one of the most important parameters in MSF process, and it is one of the few parameters that can be adjusted. It is of great significance to study the effect of recycle stream mass flowrate ($W_{Re}$) on the system performance for later control optimization research. In this paper, the recycle stream mass flowrate ($W_{Re}$) is analyzed in the range of $3 \times 10^6$ kg/h–$7 \times 10^6$ kg/h. Simulation results are shown in Figures 11–13.

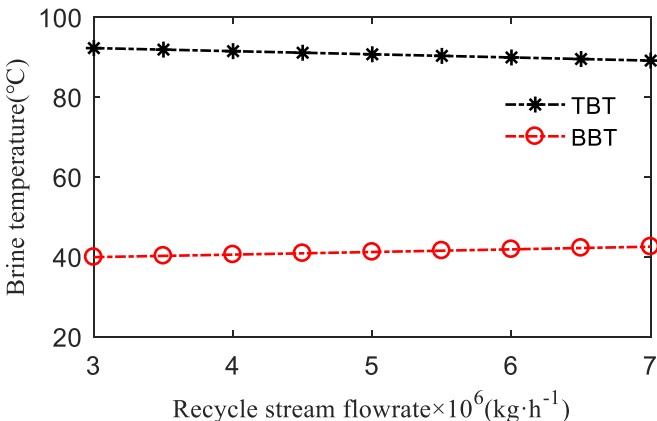

**Figure 11.** Effect of recycle stream flowrate ($W_{Re}$) on brine temperature (TBT and BBT).

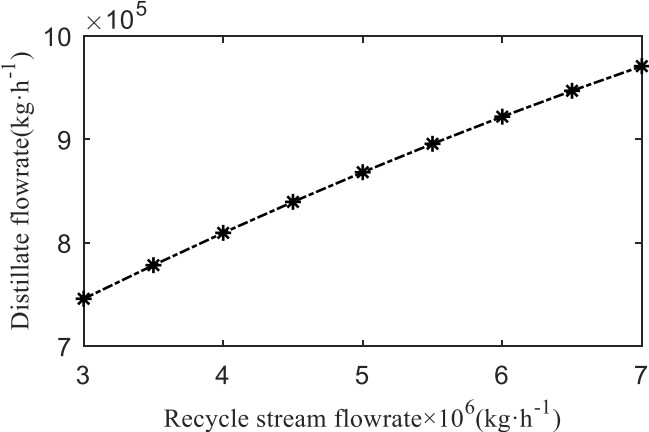

**Figure 12.** Effect of recycle stream flowrate ($W_{Re}$) on distillate flowrate ($W_{DN}$).

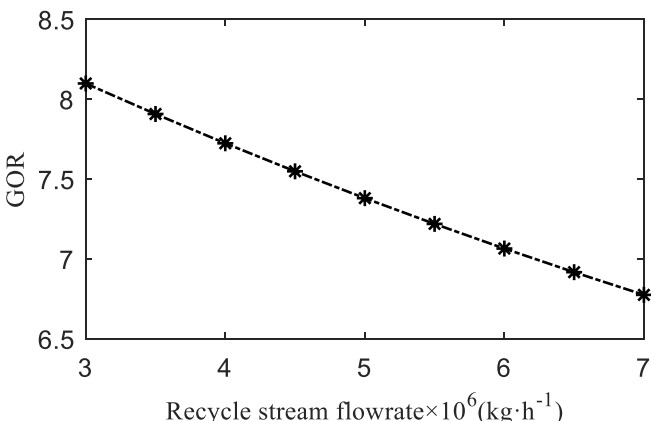

**Figure 13.** Effect of recycle stream flowrate ($W_{Re}$) on GOR.

From the simulation results, it can be seen that the increase of recycle stream flowrate ($W_{Re}$) has a small impact on TBT and BBT, but the total temperature difference between TBT and BBT shown in Figure 11 decreases quite significantly. From this perspective, the increase in recycle stream mass flowrate ($W_{Re}$) will cause a decrease in distillate production ($W_{DN}$), but the increase in recycle stream mass flowrate ($W_{Re}$) is equivalent to increasing the total brine flowrate, which will directly lead to an increase in distillate production ($W_{DN}$). It can be seen from Figure 13 that the comprehensive effect results in the increase of distillate mass flowrate ($W_{DN}$). At the same time, the increase of recycle stream flowrate ($W_{Re}$) is equivalent to the increase of brine heater load, resulting in the increase of steam consumption, resulting in the decrease of GOR, as shown in Figure 13.

## 5. Operational Optimization of MSF System

Considering that the seawater temperature, distillate demand and other operational parameters change all the time, the optimal operation of the MSF system for cost saving is quite important. Based on the established model of MSF desalination system, this paper studies the optimal operation problems with two different objectives. The first is to maximize the GOR under different operating conditions, and the second is to minimize daily operational cost in order to obtain the optimal manipulated variables at different times. Since the seawater temperature and freshwater demand are the most important and most frequently changing parameters, the optimization work selects the actual temperature from some seawater desalination plant in Zhejiang province as the background, which can be seen in Table 4. The hourly distillate demand is as shown in Table 5.

Table 4. Seawater temperature in each time period.

| Time | Temperature/°C | Time | Temperature/°C |
|---|---|---|---|
| 2:00–6:00 | 8 | 12:00–13:00 | 14 |
| 6:00–7:00 | 10 | 13:00–15:00 | 12 |
| 7:00–8:00 | 12 | 15:00–16:00 | 10 |
| 8:00–9:00 | 14 | 16:00–17:00 | 9 |
| 9:00–10:00 | 15 | 17:00–20:00 | 8 |
| 10:00–11:00 | 16 | 20:00–22:00 | 6 |
| 11:00–12:00 | 15 | 22:00–0:00 | 5 |

Table 5. Distillate demand in each time period.

| Time | Distillate Demand ($\times 10^5$ kg/h) | Time | Distillate Demand ($\times 10^5$ kg/h) |
|---|---|---|---|
| 2:00–6:00 | 6.8 | 12:00–13:00 | 8.91 |
| 6:00–7:00 | 6.88 | 13:00–15:00 | 8.79 |
| 7:00–8:00 | 7.11 | 15:00–16:00 | 8.49 |
| 8:00–9:00 | 7.53 | 16:00–17:00 | 8.11 |
| 9:00–10:00 | 8.08 | 17:00–20:00 | 8.46 |
| 10:00–11:00 | 8.72 | 20:00–22:00 | 7.29 |
| 11:00–12:00 | 9.35 | 22:00–0:00 | 6.53 |

*5.1. Optimal Operation Problem to Maximize Gained Output Ratio (GOR)*

To a large extent, the GOR is known as the traditional index to describe the performance of the MSF system. According to the flowsheet of BR-MSF shown in Figure 1, there are four manipulated variables that can be adjusted to reach the optimal operational point. These manipulated variables are recycle stream mass flowrate ($W_{Re}$), rejected seawater mass flowrate, steam temperature and reject seawater recycle mass flowrate ($C_W$). With bound constraints, given distillate demand and a well-established process model, the optimal operation problem to maximize GOR was formulated as the following problem called OPT1.

$$
\begin{aligned}
max \quad & GOR \\
s.t. \quad & f(x, u, v) = 0 \\
& W_{DN} = W_{DN}{}^* \\
& (92\ °C)T_{STEAM}^L \leq T_{STEAM} \leq T_{STEAM}^U(97\ °C) \\
& (2 \times 10^6)R^L \leq R \leq R^U(5.5 \times 10^6) \\
& (3 \times 10^6)S_w^L \leq S_w \leq S_w^U(10.2 \times 10^6) \\
& (0 \times 10^6)S^L \leq S \leq S^U(10.2 \times 10^6)
\end{aligned}
\qquad \text{(OPT1)}
$$

The superscripts $L$ and $U$ here denote the lower and upper bounds of the parameter, respectively. The equation $f(x, u, v) = 0$ represents the mechanism model of the system. Since the optimization

period is a short-term problem, the fouling factor of the brine heater and the exchanger in the flash chambers are assumed to be a constant value of $1.86 \times 10^{-4}$ (h·m$^2$·K)/kcal. At the same time, the seawater temperature and distillate demand in each time period are shown in Tables 4 and 5, respectively. The optimal operation problem was solved with interior point algorithm under the MATLAB.

The solutions including the optimal variables and the optimal objective function values are listed in Table 6.

**Table 6.** States and optimal results obtained by solving Opt1 (optimal operation problem).

| Time | $T_{sea}$ (°C) | $W_{DN} \times 10^5$ (kg/h) | $W_{Re} \times 10^6$ (kg/h) | $W_r \times 10^6$ (kg/h) | $S \times 10^5$ (kg/h) | $T_{steam}$ (°C) | GOR |
|---|---|---|---|---|---|---|---|
| 2:00–6:00 | 8 | 6.8 | 2.1972 | 7.5647 | 7.5647 | 97 | 8.9302 |
| 6:00–7:00 | 10 | 6.88 | 2.0 | 7.2253 | 7.2253 | 97 | 8.8998 |
| 7:00–8:00 | 12 | 7.11 | 2.0 | 7.0199 | 6.1930 | 97 | 8.8047 |
| 8:00–9:00 | 14 | 7.53 | 2.0 | 6.6484 | 4.2948 | 97 | 8.6357 |
| 9:00–10:00 | 15 | 8.08 | 2.0 | 6.2293 | 0.8170 | 97 | 8.4205 |
| 10:00–11:00 | 16 | 8.72 | 2.0 | 5.4808 | 0.8898 | 97 | 8.1816 |
| 11:00–12:00 | 15 | 9.35 | 2.0778 | 4.9956 | 0.5726 | 97 | 7.9565 |
| 12:00–13:00 | 14 | 8.91 | 2.0 | 5.4636 | 0.8087 | 97 | 8.1138 |
| 13:00–15:00 | 12 | 8.79 | 3.3279 | 7.0736 | 0.0829 | 97 | 8.1415 |
| 15:00–16:00 | 10 | 8.49 | 3.6552 | 7.6955 | 2.5972 | 97 | 8.2444 |
| 16:00–17:00 | 9 | 8.11 | 2.0 | 6.1698 | 6.1698 | 97 | 8.4100 |
| 17:00–20:00 | 8 | 8.46 | 3.0193 | 6.9668 | 5.5723 | 97 | 8.2683 |
| 20:00–22:00 | 6 | 7.29 | 2.5214 | 7.4993 | 7.4993 | 97 | 8.7250 |
| 22:00–0:00 | 5 | 6.53 | 2.5307 | 8.1957 | 8.1957 | 97 | 9.0367 |

Having assigned the steam temperature ($T_{steam}$) of the system, the seawater temperature ($T_{sea}$) and distillate demand ($W_{DN}$) for each time period, for the Opt1 problem that maximizes the GOR, optimal operations of the series of the recycle stream mass flowrate ($W_{Re}$), mass flowrate to the reject seawater splitter ($W_r$) and reject seawater recycle mass flowrate ($S$) were obtained. Here, note that the equipment is not working during maintenance from 00.00 to 02.00 at night.

The overall optimization results also show that lower seawater temperature ($T_{sea}$) can result in higher GOR. During the time period of 22:00–00:00, the seawater temperature shows the lowest value in a day. At the same time, the reject seawater recycle mass flowrate ($S$) reaches a maximum value, and the GOR reaches a peak value, which is to compensate for the effect of the low seawater temperature on the system performance. It is also found that, at relatively high temperature and high distillate demand, the value of S is quite small. This means in the conditions that the discharged brine from rejection section should not be reused and mixed with origin feed seawater.

Figure 14 shows the relationship of distillate demand and rejected recycle mass flowrate ($S$), from which a conclusion can be drawn. Figure 15 shows the profile of optimal the recycle stream mass flowrate ($W_{Re}$) and the reject seawater splitter ($W_r$) under the given ambient parameter change to maximize the GOR.

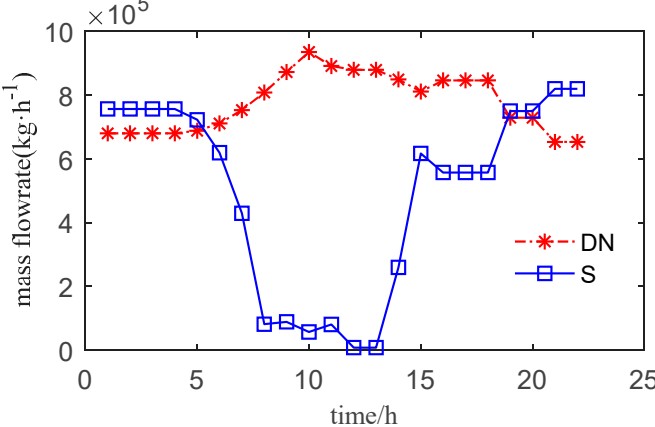

**Figure 14.** Trend chart of optimal operation parameters aiming at maximum GOR.

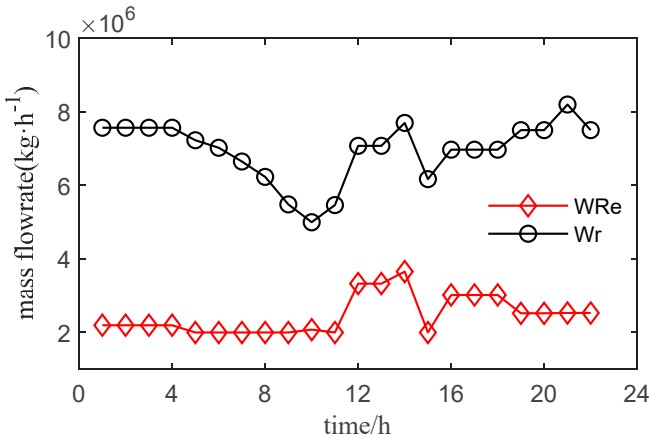

**Figure 15.** Trend chart of optimal operation parameters aiming at maximum GOR.

### 5.2. Optimal Operation Problem to Minimize Daily Operational Cost

The production cost is directly affected by the higher operating cost of the MSF desalination system. The electricity price, seawater temperature etc. change frequently over time, causing the operation control difficulty of MSF system. Therefore, the optimal operation of the MSF system was further studied below. The optimal operation problem to minimize the daily operational cost can be formulated as:

$$
\begin{aligned}
&min \quad TOC \\
&s.t. \quad f(x, u, v) = 0 \\
&\qquad TBT = TBT^* \\
&\qquad W_{DN} = W_{DN}^* \\
&\qquad (92\,^\circ\text{C})T_{STEAM}^L \le T_{STEAM} \le T_{STEAM}^U (115\,^\circ\text{C}) \\
&\qquad (2 \times 10^6)R^L \le R \le R^U (5.5 \times 10^6) \\
&\qquad (3 \times 10^6)S_w^L \le S_w \le S_w^U (10.2 \times 10^6) \\
&\qquad (0 \times 10^6)S^L \le S \le S^U (10.2 \times 10^6)
\end{aligned}
\tag{OPT2}
$$

Here $W_{DN}$ is the total distillate production, $W_{DN}^*$ is the given distillate demand, which is shown in Table 5, and $TBT^*$ is expected value of $TBT$. $TOC$ denotes the total operational cost of each hour, which can be calculated as follows:

$$
TOC = TOC1 + TOC2 + TOC3 + TOC4 + TOC5 \tag{33}
$$

$TOC1$ denotes steam cost, $TOC2$ denotes pump energy consumption, $TOC3$ denotes chemical additive cost, $TOC4$ denotes labor cost, $TOC5$ denotes system maintenance and servicing costs. The electricity price of each time period is shown in Figure 16. The steam price at 92–115 °C is 23.72 CNY/Million kJ.

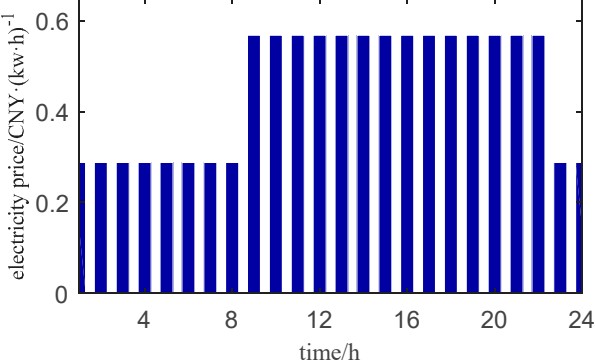

**Figure 16.** Changes in electricity prices within a day.

Each part of the operational cost is as follows:

$$TOC1 = W_{steam} \times h_V \times s_P \tag{34}$$

$$TOC2 = e_P \times P \times T \tag{35}$$

where, $P = \frac{g \cdot Q \cdot H}{1000 \times 3600 \cdot \eta} \times 1.2$.

$$TOC3 = \frac{WR}{\rho_B} \times 0.11 \tag{36}$$

$$TOC4 = \frac{DN}{\rho_D} \times 0.694 \tag{37}$$

$$TOC5 = (TOC1 + TOC2 + TOC3 + TOC4) \times 0.03 \tag{38}$$

With the operational cost equations and the established MSF system model, the optimal operation problem named as OPT2 was studied and solved by an interior point algorithm under the MATLAB platform. The problem takes into account the changes of the seawater temperature ($T_{sea}$), distillate demand and electricity price at different hours in a day. The problem was successfully solved and the key results are listed in Table 7. Since the optimization results show that the optimal reject seawater recycle mass flowrate ($S$) is 0 at each time interval, it is not included in Table 7 for convenience.

**Table 7.** States and optimal results obtained by solving Opt2.

| Time | $Tsea$ (°C) | $W_{DN} \times 10^5$ (kg/h) | $W_{Re} \times 10^6$ (kg/h) | $W_r \times 10^6$ (kg/h) | $T_{steam}$ (°C) | $W_{steam} \times 10^4$ (kg/h) | TOC (CNY/h) |
|---|---|---|---|---|---|---|---|
| 2:00–6:00 | 8 | 6.8 | 4.0271 | 9.4133 | 92.5587 | 7.9868 | 6669 |
| 6:00–7:00 | 10 | 6.88 | 4.0740 | 9.3036 | 92.6141 | 8.0848 | 6754 |
| 7:00–8:00 | 12 | 7.11 | 4.2170 | 9.1338 | 92.7676 | 8.4213 | 7024 |
| 8:00–9:00 | 14 | 7.53 | 4.4895 | 8.8870 | 93.0528 | 9.0742 | 7540 |
| 9:00–10:00 | 15 | 8.08 | 4.3540 | 8.1832 | 93.4182 | 9.9298 | 8580 |
| 10:00–11:00 | 16 | 8.72 | 4.6610 | 7.7720 | 93.8824 | 11.0101 | 9424 |
| 11:00–12:00 | 15 | 9.35 | 4.8472 | 7.4242 | 94.3241 | 12.0852 | 10257 |
| 12:00–13:00 | 14 | 8.91 | 4.7363 | 7.8370 | 93.9868 | 11.3113 | 9647 |
| 13:00–15:00 | 12 | 8.79 | 4.7571 | 8.1304 | 93.8761 | 11.0991 | 9469 |
| 15:00–16:00 | 10 | 8.49 | 4.6603 | 8.4505 | 93.6492 | 10.6028 | 9072 |
| 16:00–17:00 | 9 | 8.11 | 4.5134 | 8.7067 | 93.3862 | 9.9916 | 8592 |
| 17:00–20:00 | 8 | 8.46 | 4.7148 | 8.6449 | 93.6173 | 10.573 | 9036 |
| 20:00–22:00 | 6 | 7.29 | 4.1718 | 9.2186 | 92.8529 | 8.7493 | 7614 |
| 22:00–0:00 | 5 | 6.53 | 3.9008 | 9.6128 | 92.3985 | 7.6395 | 6385 |

After assigning the seawater temperature ($T_{sea}$) and distillate demand ($W_{DN}$) for each time period, for the Opt2 problem that minimizes the TOC, the optimal operations of the series of the recycle stream mass flowrate ($W_{Re}$), mass flowrate to the reject seawater splitter ($W_r$), Steam temperature ($T_{steam}$) and Steam mass flowrate ($W_{steam}$) were obtained. Here note that the equipment is not working during the maintenance from 0 to 2 o'clock at night.

The overall optimization results also show that from two o'clock in the morning, the seawater temperature ($T_{sea}$) gradually increases, and the demand for distillate ($W_{DN}$) also increases. Therefore, higher flash brine volume and steam consumption ($W_{steam}$) are required, resulting in higher operating costs (TOC). During the period of 11:00–12:00, the demand for distillate ($W_{DN}$) reaches the peak in a day, and the temperature of seawater ($T_{sea}$) is also at a high level. At the same time, the operating cost (TOC) of the device also reaches the maximum in this period. After 12:00, the temperature of seawater ($T_{sea}$) gradually decreases, and the distillate demand ($W_{DN}$) maintain a downward trend from 12:00 to 17:00, while it increased from 17:00 to 20:00, and began to decline after 20:00. The operating cost of the device is the same as the trend of distillate demand ($W_{DN}$).

Figure 17 shows the relationship between TOC and steam temperature ($T_{steam}$) and steam flow ($W_{steam}$) in each time period. Figure 18 shows the profile of optimal the recycle stream mass flowrate ($W_{Re}$) and the reject seawater splitter ($W_r$) under the given ambient parameter change to minimize the TOC.

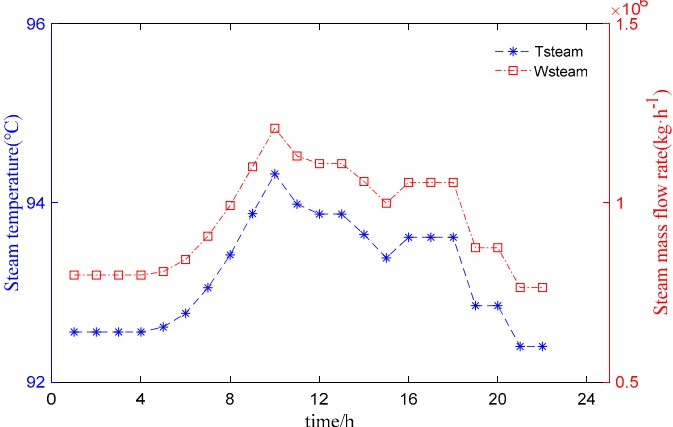

**Figure 17.** Trend of optimal operating parameters aiming at minimum TOC.

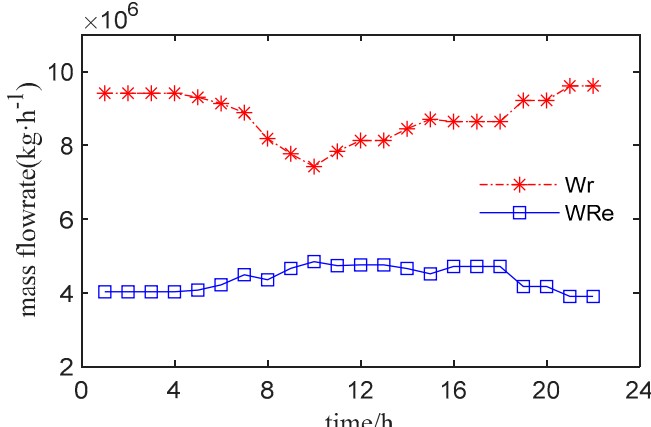

**Figure 18.** Trend of optimal operating parameters aiming at minimum TOC.

## 6. Conclusions

The multi-stage flash desalination process is one of the most important technologies to obtain fresh water on a large scale. However, its operational cost is relatively high and its performance is significantly affected by seawater temperature, salt concentration, steam quality and other operational factors. In this paper, the sensitivity analyses of these parameters on the performance of MSF system were carried out based on the elaborated established process model, and then two kinds of operational optimization problems were studied.

To obtain a more comprehensive and more elaborate model, this paper considers the effects of factors such as boiling point elevation, influence of heat loss of condenser tubes, the unequal temperature drops in each flash stage, and effect of the reject seawater recycle mass flowrate. Simulation results of an MSF system with a 16-stage flash chamber demonstrate that the model has similar accuracy with those from Rosso and Mujtaba. The sensitivity analysis of seawater temperature shows that higher seawater temperature can cause higher BBT and lower distillate flowrate, and there is a 'best temperature' to maximize GOR (gained output ratio). If the feed seawater is quite low, the increase of reject seawater recycle mass flowrate(S) will decrease the distillate flowrate, but will increase the GOR to a maximum until the mixed seawater reach the 'best temperature'. Steam temperature is the only factor which has a significant and positive effect on distillate flowrate and GOR.

The optimal operation of MSF was studied with two different objective functions. The optimization problem to maximize the GOR under given daily seawater temperature and fresh water demand shows that with limited upper steam temperature of 97 °C, the maximum GOR can be achieved with given demand and feed seawater temperature. However, at a high freshwater demand time interval, the rejected seawater recycle mass flowrate(S) is quite large, so the rejected seawater recycle mass flowrate should not be ignored and set to zero. Also, the recycle stream mass flowrate and the rejected seawater mass flowrate should be adjusted as ambient conditions change. To minimize the daily operational cost, the objective function was reformulated considering the pump power cost of other mass flowrate and the price of steam in China. Computing results of the optimization shows that, with the same seawater and freshwater demand, the optimal operation is quite different from that from OPT1. Optimal value of the rejected seawater recycle mass flowrate(S) is always zero, the steam flowrate and temperature increase as the demand for freshwater increase, but the rejected seawater mass flow decreases at the same time. With the established process model and objective function, the optimal values for those manipulated variables can be obtained, which is helpful to guide the economical operation of the MSF system.

**Author Contributions:** H.G. and Q.H. performed the simulations and analyzed the data, A.J. designed the process scheme and optimization of the paper, Q.H., Y.X. and J.W. wrote the paper and reviewed it, F.G. checked the results of the whole manuscript. All authors have read and agreed to the published version of the manuscript.

**Funding:** The work was supported by the Natural Science Foundation of Zhejiang No. (LY20F030010, LQ19E060007), the National Natural Science Foundation of China (No. 61973102) and the National Science and Technology Major Project (2018AAA0101601).

**Conflicts of Interest:** The authors declare no conflict of interest.

## Abbreviations

In order to better understand the mathematical model of MSF seawater desalination, the symbol description list is added here.

| | |
|---|---|
| $A_H$ | Heat transfer area of the brine heater, m$^2$ |
| $A_j$ | Heat transfer area of stage $j$, m$^2$ |
| $BPE_j$ | Boiling point elevation of stage $j$ |
| $C_W$ | Rejected seawater mass flowrate, kg·h$^{-1}$ |
| $C_F$ | Feed seawater salt concentration, wt% |
| $C_{B0}$ | Salt concentration in the flashing leaving the brine heater, wt% |
| $C_{Bj}$ | Salt concentration in the flashing brine leaving stage $j$, wt% |
| $CP_{Bj}$ | Heat capacity of brine leaving stage $j$, kcal·(kg·°C)$^{-1}$ |
| $CP_{Dj}$ | Heat capacity of distillate leaving stage $j$, kcal·(kg·°C)$^{-1}$ |
| $C_m$ | Salt concentration in make-up water, wt% |
| $CP_{Rj}$ | Heat capacity of cooling brine leaving stage $j$, kcal·(kg·°C)$^{-1}$ |
| $CP_{RH}$ | Heat capacity of cooling brine leaving brine heater, kcal·(kg·°C)$^{-1}$ |
| $C_R$ | Salt concentration in the cooling brine to the recovery section, wt% |
| $C_{Re}$ | Recycle brine concentration, wt% |
| $D_H^i$ | Internal diameter of condenser tube, m |
| $D_H^o$ | External diameter of condenser tube, m |
| $D_j^i$ | Internal diameter of condenser tube at stage $j$, m |
| $D_j^o$ | External diameter of condenser tube at stage $j$, m |
| $e_P$ | Electricity price, CNY/h |
| $f_{BH}$ | Brine heater fouling factor, h·m2·°C·kcal$^{-1}$ |
| $f_j$ | Fouling factor at stage $j$, h·m2·°C·kcal$^{-1}$ |
| $GOR$ | Gained output ratio |
| $H$ | Pump stroke, m |
| $h_{Bj}$ | Specific enthalpy of flashing brine at stage $j$, kcal·kg$^{-1}$ |
| $h_{Dj}$ | Specific enthalpy of distillate at stage $j$, kcal·kg$^{-1}$ |
| $h_{Re}$ | Specific enthalpy of recycle stream at stage $j$, kcal·kg$^{-1}$ |

| | |
|---|---|
| $h_m$ | Specific enthalpy of make-up brine at stage $j$, kcal·kg$^{-1}$ |
| $h_R$ | Specific enthalpy of stream to recovery section, kcal·kg$^{-1}$ |
| $h_S$ | Specific enthalpy of recycle brine at rejection stage, kcal·kg$^{-1}$ |
| $h_{Vj}$ | Specific enthalpy of steam at stage $j$, kcal·kg$^{-1}$ |
| $h_{W_F}$ | Specific enthalpy of brine at the entrance of rejection section, kcal·kg$^{-1}$ |
| $h_{W_S}$ | Specific enthalpy of feed seawater, kcal·kg$^{-1}$ |
| $H_j$ | Height of condenser tube at stage $j$, m |
| $L_H$ | Length of brine heater condenser tube, m |
| $L_j$ | Length of condenser tube at stage $j$, m |
| $N$ | Total number of stages, N = NR + NJ |
| $NETD$ | Non-equilibrium allowance, °C |
| $NJ$ | Number of stages in the heat rejection section |
| $NR$ | Number of stages in the heat recovery section |
| $S_P$ | Steam price, CNY/MkJ |
| $Q$ | Mass flow, kg/h |
| $S$ | Reject recycle mass flowrate, kg·h$^{-1}$ |
| $T_{Bj}$ | Temperature of flashing brine leaving stage $j$, °C |
| $T_{B0}$ | Temperature of flashing brine leaving the brine heater, °C |
| $T_{Dj}$ | Temperature of distillate leaving stage $j$, °C |
| $T_{Fj}$ | Temperature of cooling brine leaving stage $j$, °C |
| $T_{F0}$ | Temperature of cooling brine to brine heater, °C |
| $TL_j$ | Temperature loss due to demister and condenser, °C |
| $T_{Vj}$ | Temperature of flashed vapor at stage $j$, °C |
| $T_{sea}$ | Seawater temperature, °C |
| $T_{steam}$ | Steam temperature, °C |
| $U_H$ | Overall heat transfer coefficient at the brine heater, kcal·(m$^2$·h·°C)$^{-1}$ |
| $U_j$ | Overall heat transfer coefficient at stage $j$, kcal·(m$^2$·h·°C)$^{-1}$ |
| $V_{Bj}$ | Evaporation capacity of brine at stage $j$, kg·h$^{-1}$ |
| $W_{B0}$ | Flashing brine mass flowrate leaving brine heater, kg·h$^{-1}$ |
| $W_{BD}$ | Blowdown mass flowrate, kg·h$^{-1}$ |
| $W_{Bj}$ | Flashing Brine mass flowrate leaving stage $j$, kg·h$^{-1}$ |
| $W_{BN}$ | Flashing Brine mass flowrate leaving stage N, kg·h$^{-1}$ |
| $W_{Dj}$ | Distillate mass flowrate leaving stage $j$, kg·h$^{-1}$ |
| $W_{DN}$ | Distillate mass flowrate leaving stage N, kg·h$^{-1}$ |
| $W_F$ | Flashing seawater mass flowrate to rejection section, kg·h$^{-1}$ |
| $W_j$ | Width of condenser tube at stage $j$, m |
| $W_m$ | Make-up brine mass flowrate, kg·h$^{-1}$ |
| $W_R$ | Cooling brine mass flowrate to recovery section, kg·h$^{-1}$ |
| $W_r$ | Mass flowrate to the reject seawater splitter, kg·h$^{-1}$ |
| $W_{Re}$ | Recycle stream mass flowrate, kg·h$^{-1}$ |
| $W_S$ | Seawater mass flowrate, kg·h$^{-1}$ |
| $W_{steam}$ | Steam mass flowrate, kg·h$^{-1}$ |
| $\lambda_s$ | Latent heat of steam, kcal·kg$^{-1}$ |

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
