# Peer review of "Mode-Based Analysis and Optimal Operation of MSF Desalination System"

_processes, doi:10.3390/pr8070794_

Round 1

Reviewer 1 Report

This is a good article, well written with suitable scientific rigor. The paper is publishable, but there are some minor revisions that need to be made. see attached for details.

Author Response

Thank you very much for giving us your suggestions and help us correct the syntax errors. You are the best and the most patient reviewer I met. With your help, we correct the errors and check the manuscript again. We wanted to turn to editorial services to make English more readable, but since we made more revision in the part of introduction and simulation results, the time is limited. So if the english is still not good for publish and more time is given to us, we will continue improve our manuscipt. Again, thank you vevy much.

Reviewer 2 Report

  • In this article, we have worked with a model of the multi-stage flash desalination system and studied the effects of different operating parameters on system performance and optimization to save costs.

  • In the introduction, the necessary articles and previous works are indicated. The modeling has focused on steady-state mathematical using papers [7] and [8]. Bibliographic references are old and focused on the first decade. An update of the references and incorporating dynamic modeling of the MSF systems would be convenient.

  • In section 4.2 with its subsections, the graphic results of the variations the feed seawater temperature, the reject seawater recycle mass flowrate, the steam temperature, and the recycle stream mass flowrate on system performance are presented. From lines 204-298, section 4.2, it is excessive due to the number of exposed figures and the GOR and distilled water resolutions question the model and its precision.

  • The sea water temperature ranges will be from 26ºC to 46ºC (line 207), as it affects the precision of the model used. However, the values for the seawater temperature in section 5 are from 8ºC to 16ºC (Table 4). As these temperature differences affect the model, it has been validated ...

  • Important changes in the demand for distilled water are not justified (6.8-9.35 105 kg / h, table 5)

  • The two optimization programs Max GOR and Min TOC have been developed to achieve the objective functions. Using the Matab for this purpose. The procedure is simple and not all variables have been incorporated:

Fouling factor,

validation of the model for dynamic situations,

How it affects the expected changes (Seawater temperature) in the GOR and in the distilled water over time

How long it takes for the system to stabilize all its parameters against a change.

  • Increase the size of the appendix symbols

Author Response

Thank you very much for giving us your cordial suggestions. According to your suggestions, we revised our introduction, and in this part, we added some new research literatures. As you mentioned, we also present the studies of the dynamic modeling and optimization. Frankly speaking, we also did some work in dynamic modeling and simulation. But I think what we did need improvement as present time. So, we focused on the optimal operation of MSF process in this manuscript, and considered the sea water temperature and other conditions in our region. And since many researchers established the dynamic equations for estimation of fouling resistance, and the fouling factor changes slowly in short term. So, in our work, the fouling factor was used as constant value in the operational optimization problem. If we want to optimize the process for medium and long term, we should predict or get the value by dynamic model or parameter identification. From the literature and our simulation results, if the dynamic conditions don't fluctuate in large range, the transient process is not long, generally within several minutes to half an hour. More revision and respond can be seen in the attachment.

Round 2

Reviewer 2 Report

The authors have made numerous changes to the article and have made the current version a complete paper. The work is simulation and optimization based on simulation, therefore it is not original and its contribution is to link the simulation with optimization processes.

The methodology and steps carried out are appropriate and its reading is easy and comfortable. The size of the work is very extensive, and can be reduced without much difficulty, although the authors have chosen to show too much detail, especially in the figures included.